# Investigation of the Effects of Polymer-Based Grinding Aids on the Surface Chemistry Properties of Cement

**DOI:** 10.3390/polym17192691

**Published:** 2025-10-04

**Authors:** Kenan Çinku, Ebru Dengiz Özcan, Şenel Özdamar, Hasan Ergin

**Affiliations:** 1Department of Mining Engineering, Istanbul University—Cerrahpaşa, Istanbul 34500, Türkiye; kenan@iuc.edu.tr; 2Department of Mining Engineering, Istanbul Technical University, Istanbul 34469, Türkiye; hergin@itu.edu.tr; 3Department of Geology Engineering, Istanbul Technical University, Istanbul 34469, Türkiye; ozdamarse@itu.edu.tr

**Keywords:** polymer, zeta potential, grinding aid, dispersion, turbidity, ion chromatography, rheometer

## Abstract

Polymer-based superplasticizers represent an emerging class of additives in cement and concrete production with demonstrated effects on zeta potential, ion exchange, turbidity and rheological behavior during hydration. This study examines the influence of polymer-based grinding aids focusing on the dosage of A2 on the grinding performance of Portland cement. Among the tested additives, A2 exhibited superior dispersing ability and agglomeration-preventing activity, yielding a zeta potential of −8.98 mV. Correspondingly, the release of the ion concentration of Ca^2+^ decreased to 190 mg/L, while SO_4_^2−^ increased to 400 mg/L, indicating enhanced ionic interaction at the optimal A2 dosage of 2.5 g. The turbidity tests further revealed that cement samples ground with 2.5 g of A2 remained homogeneously suspended for longer periods compared to other additives. Overall, the analysis of cement surface properties confirmed that polymer-based grinding aids, particularly A2, significantly improve the dispersion stability of cement particles during grinding.

## 1. Introduction

Polycarboxylate-based superplasticizers (PCEs) are a significant type of concrete admixture that have been used in many engineering projects. PCEs are key component of concrete, offering notable design flexibility and high performance due to their molecular structure. Performance can be tuned by adjusting and optimizing parameters such as backbone length, side chain length and functional groups [1,2,3,4].

PCE macromolecules have a specific function in concrete and cement science: they address key challenger identified in these fields. Given the significance of this phenomenon, it is imperative to investigate and comprehend the dispersion behavior of cement particles. The addition of PCEs to cement particles disperses into smaller agglomerates, which become the predominant entities of the cement paste in the concrete mixture. This dispersion has a significant impact on the fluidity and workability of concrete. The development of a negative charge on the cement particles develops as a consequence of this process [5,6]. Van der Waals forces, which appear to control agglomeration, can be neutralized by anionic polymers adsorbed onto the particles, as well as by sulphonic groups on their surface [7]. Cement particle dispersion can be attributed to electrostatic repulsion arising from the adsorption of negatively charged groups. These forces are chiefly contingent on the extent to which PCEs are adsorbed onto the cement particle surface [8]. Consequently, the magnitude of PCE adsorption on the cement particle surface directly correlates with the magnitude of repulsive forces between particles, thereby affecting the fluidity of the cement paste [9]. Therefore, an understanding of the behavior of cement particles can clarify the interaction between cement and PCEs, which in turn helps elucidate the adsorption behavior [10,11,12,13,14].

In order to characterize the interaction between cement and PCEs, various researchers [15,16] have studied the zeta potentials generated at the interface between cement particle surfaces and the PCEs. The action of first- and second-generation superplasticizers is based on electrostatic repulsion. Figure 1 shows the mechanism of superplasticizers in cement paste. In the presence of superplasticizers (SP), the air content tends to decrease because negatively charged cement particles, result from the adsorption of SP molecules. Consequently, there is less space for air bubbles to be entrapped. The negative charge also keeps the cement particles and air bubbles dispersed and reduces the friction of the mixture. The action of the polycarboxylate ether (PCE)-based superplasticizer causes a spatial (steric) separation of the cement particles, as shown on the right of Figure 1. PCEs have a specific number of side chains distributed along a main chain. The carboxylate groups interact with the surface of the cement grains, leading to polymer adsorption. Steric repulsion is considered a primary mechanism for dispersing cement grains [17,18].

The adsorption behavior of various additives, including viscosity modifiers, water-reducing admixtures and air-entraining agents (AEAs), has been identified as a primary factor influencing their efficacy in the context of cement and concrete [19,20,21]. Most of these additives are surfactants, defined as compounds with hydrophobic or nonpolar chain tails and one or more hydrophilic headgroups [22].

A surfactant’s headgroup can be categorized into one of three types: anionic, cationic or nonionic. Several studies have indicated that cement particles can possess a positive electrical charge [5,9,23]. However, numerous studies have provided ample evidence that cement particles can be negatively charged [6], and that anionic organics such as superplasticizers, cellulose and latex can be adsorbed by negatively charged cement particles [24,25]. Research has shown that both cationic and anionic surfactants can be adsorbed by cement particles [26]. In light of discrepancies observed in the aforementioned investigations, most researchers have proposed a mosaic structure for hydrating cement particles characterized by heterogeneous charge distribution [7,24,25,27,28]. Accordingly, this study characterizes and examines anionic polymer-based surfactants with respect to the zeta potential, ion chromatography and turbidity behaviors of cement particles.

## 2. Materials and Methods

### 2.1. Materials

In this study, Portland Cement CEM I 42.5 R, consisting of 91% clinker, 4% gypsum and 5% limestone mixture according to EN 197-1 European Cement Standard, was supplied by Nuh Cement Factory, Istanbul, Türkiye [29]. As shown in Table 1, geochemical characterization of the raw materials was performed using X-ray fluorescence (XRF) Bruker S8 TIGER WDXRF Spectrometer (Bruker AXS GmbH, Karlsruhe, Germany; operated with SPECTRAplus software, version 2.3) and the clinker’s mineralogy was analyzed using X-ray diffraction (XRD) Bruker D8 ADVANCE Diffractometer (Bruker AXS GmbH, Karlsruhe, Germany; equipped with DIFFRAC.SUITE software, version 1.9), as illustrated in Figure 2. 

The XRD patterns results of the studied clinker indicate that C_2_S exhibits a monoclinic crystal structure, while C_3_A and C_3_S display cubic symmetry and C_4_AF features an orthorhombic configuration (Figure 2). The following primary aluminum oxide (Al_2_O_3_) values are of particular interest: 29.3, 30.07, 32.1, 32.7, 34.2. In addition, the secondary aluminum oxide (Al_2_O_3_) values of particular relevance are 31.07 (ß), 33.03, and the aluminum (Al) values of 20.96, 21.77, 33.18, 33.25. Finally, the ferrite values of particular note are 12.20, 24.35, 32.12, and 33.88.

In addition to the XRD and XRF analyses of the clinker prior to the grinding process, the 32 µm (%) sieve residue and Blaine (cm^2^/g) values were also examined. In the laboratory-type grinding mill, the mill was stopped every 10 min for a total of 60 min, after which the 32 µm (%) sieve residue and Blaine (cm^2^/g) values of the samples taken homogeneously from each part of the mill were measured. As illustrated in Figure 3, the mathematical correlation curve of these data is evident.

The sieve residue of the clinker at 32 µm is 16.1% (%). Since the laboratory conditions are intended to be identical to the operational conditions of Nuh Cement Factory, Istanbul, Türkiye, the clinker was subjected to pre-crushing, resulting in a d_80_ size of 1.5 mm.

### 2.2. Characterizations of PCEs

Six types of admixtures (TEA-TIPA-AMA/6E-A1-A2-A3) used as water reducers and strength-enhancing agents, were added to the grinding process one by one in order to investigate their effects. The chemicals A1, A2 and A3 were synthesized from a VPEG-based polymer by radical polymerization in and their general formulas are given in Figure 4.

It is evident that R1, R2, and R3 each represent a H atom or a methyl group. R4 is indicative of either a H atom or 1–10 hydrocarbon groups. RO represents an alkaline group containing between 2 and 5 carbons. The symbol ‘m’ is used to denote the number of moles of ‘RO’, while ‘x’ represents the alkaline group containing 1–5 carbons. The chemicals A1-A3 were produced at 30 °C, with A1-A3 being produced in 2 h while A2 required in 3 h and 40 min. Vinylpolyethyleneglycol (VPEG) was synthesized using acrylic acid, a copolymer. This is an addition reaction initiated by a free radical starting chemical. Azobisisobutyronitrile (C_8_H_12_N_4_) was used as an initiator. It works at low temperatures. It is an organic solvent. Their chemical and physical properties are presented in Table 2. AMA-6E is a commercial product produced by a factory of French origin and sold in Turkey as a cement grinding chemical. In the cement industry, TEA and TIPA are frequently utilized as grinding aids [30].

### 2.3. Methods

#### 2.3.1. Zeta Potential

Six types of grinding aids (TEA-TIPA-AMA/6E-A1-A2-A3) and six different dosages of A2 polymer were utilized in grinding tests. In the present study, the investigation focused on thirteen cement samples, utilizing a zeta-potential analysis approach. Cement samples were subjected to grinding with various chemicals, and the resulting powder was transferred into 50 mL Falcon tubes.

The sample preparation for zeta potential measurements was conducted in three stages. In the initial phase of the experiment, 49 mL of water and 1 g of chemical agent-aided cement were separately introduced into a 50 mL Falcon tube. These substances were then agitated using a shaker for 15 min to achieve homogeneous suspensions. Subsequently, these suspensions were subjected to a centrifugal process at 6000 rpm for 5 min to ensure phase separation. After the centrifugation, the supernatant solution was carefully transferred to an additional 10 mL Falcon tube. Thereafter, 0.1 mg of the cement grains that had settled during the second stage of the procedure were added using a micro spatula, followed by vigorous shaking 10 times. Samples were placed in the NanoBrook ZetaPALS (Brookhaven Instruments Corp. Nashua, NH, USA) cell at 1.5 mL, and seven measurements were taken, each for three cycles. Measurements were conducted again the following day under the same conditions to ensure maximum data reliability. The generation of graphs was achieved by the calculation of the mean value of the data. The PALS method is a process in which a particle is required to move a distance that is only a fraction of its diameter. This method is considered to be ideal for samples that exhibit very low mobility. The phenomenon of low mobility may be attributed to various factors, including the presence of particles dispersed in oily or unusually viscous media, the existence of samples with weak charges near their isoelectric points, or the generation of a field strength of 1–2 V/cm in certain samples.

#### 2.3.2. Turbidity

Turbidity is defined as a measure of the level of particles, such as sediment, plankton, or organic by-products in a water body. As turbidity levels in water increase, the water’s density and clarity decrease. This is due to an increase in the concentration of light-blocking particles. Turbidity meters are instruments that employ a combination of a light source and a photo detector to quantify light scattering phenomena. The resulting data are expressed in turbidity units of measurement, including nephelometric turbidity units (NTU) and formazan turbidity units (FTU). In this study, the Thermo Orion AQUAfast II AQ2010 (Thermo Fisher Scientific Inc., Beverly, MA, USA) turbidity instrument was utilized. Thirteen samples were prepared accordingly. The sample preparation process was divided into two stages. The initial phase of sample preparation involved the utilization of three distinct grinding chemicals (namely, TEA, TIPA, and AMA-6E) in conjunction with a reference sample. These samples were then placed sequentially into 50 mL Falcon tubes, with a volume of 49 g of water per 1 g of chemical agent-aided cement sample. The samples were then subjected to a shaking process by hand for a duration of 15 min. Following this, the samples were subjected to a centrifugation at 6000 rpm for a duration of 5 min. The aliquot located at the superior extremity of the centrifuge samples was separated and transferred to the turbidity test cell, following which the experiment was conducted.

The initial sample (n = 13) was subjected to a centrifugal process for a duration of one minute with the objective of measuring turbidity. At this juncture, the turbidity sample was transferred into a Falcon tube, and the turbidity was subsequently measured by selecting the appropriate measurement mode. Subsequently, the sample was measured at intervals of 5 min, 15 min, 30 min, 1 h, 12 h and 24 h.

#### 2.3.3. Ion Chromatography

The samples prepared for turbidity measurements were prepared using the same method of preparation and placed in Falcon tubes. The Thermo Dionex ICS-1100 (Thermo Fisher Scientific Inc., Beverly, MA, USA) was utilized for the measurement of ion chromatography. The liquid samples are subjected to filtration to remove sediment and other particulate matter and to limit the potential for microbial alteration before analysis. A series of rinses was performed on the aqueous samples with sample water, utilizing a sterile syringe to ensure sterility. The samples were then subjected to filtration through 0.45 µm filters, a process that enabled the removal of any particulate matter. The collection bottle was also subjected to three rinses with filtrate before being filled with sample filtrate. The 5 mL sample required for analysis was then transferred into the instrument. Ion chromatography is a method of separating ions based on their interaction with a stationary phase, which is composed of a resin, and a mobile phase, which is composed of an eluent. The phases differ between an anion column, which attracts anions, and a cation column, which attracts cations. It is important to note that each column is only able to measure the conductivity of the specific type of ion to which it is attracted. The ions will move at different speeds through the columns of the ion chromatograph depending on their affinity for the specific resin and are separated from each other based on differences in ion charge and size. As the eluent passes through the column, ions with weaker binding affinity for the resin move more rapidly through the column and are thus separated first, while ions with stronger binding affinity for the resin move more slowly through the column. After exiting the column, the ions are measured with an electrical conductivity detector. The detector produces a chromatogram, which is a graphical representation of the conductivity over time. Each ion produces a peak on this graph, the height of which is contingent on the relative ion concentration in the injected solution.

#### 2.3.4. Rheology

The samples were prepared for viscosity measurements using Brookfıeld R/S Plus Rheometer (Brookfield Engineering Laboratories, Inc., Middleboro, MA, USA). It is an indisputable fact that cement mortars are viscous fluids [31,32]. The test procedure is selected according to the material properties under investigation, for example, constant flow, thixotropic, or visco-elastic behavior in rotational or oscillatory tests. It is clear that the constant flow properties of cement mortars are related to viscosity. Cement samples were obtained from raw materials subjected to the grinding process with different grinding chemicals. The samples were prepared at a concentration of 15% [solute (15 g cement)/solute (85 g water)] and viscosity measurements were made using a rheometer.

## 3. Results and Discussion

To achieve a more profound analysis of the chemistry of the cement surface, a series of investigative procedures was carried out. Thirteen cement samples were tested in this study. They were coded as reference, Tea, Tıpa, AMA-6/E, A1-2.5 g, A2-0.5 g, A2-1 g, A2-1.5 g, A2-2 g, A2-2.5 g, A2-3 g, A2-4 g, A3-2.5 g. The tests are listed in Table 3.

The results of the analyses were examined in order to investigate the effects of amine group and polymer group grinding chemicals on cement grains, and to obtain information about their mechanisms of action.

In order to further investigate the electrokinetic characteristics, the zeta potential of cement paste containing a synthesized water-reducing and strength developer type of PCE was measured. The zeta potential results, with varied PCE dosages ranging from 0% to 4%, are illustrated in Figure 5 and Figure 6. The zeta potential of cement paste initially exhibits a positive value, but then changes to a negative value after adding the synthesized PCE, owing to the adsorption of anionic groups of the synthesized PCE on the surfaces of cement particles. It has been shown that there is a direct correlation between the PCE dosage and the increase in zeta potential. This is attributable to the replacement of high-valent anions by adsorbed anionic groups of PCE on the surfaces of cement particles. This is an example of an unequal ion exchange process known as competitive adsorption [31,32,33,34,35,36].

The high absolute value of zeta potential is indicative of the high intensity of anionic charge provided by the added PCE, which further illustrates the high adsorption amount on cement particles. Consequently, electrostatic repulsion between cement particles increases, leading to the release of free water from cement aggregates. This, in turn, results in an increased dispersion capacity for the synthesized PCE, which reduces viscosity. This is in accordance with the previously mentioned fluidity data.

An analysis of the zeta potential values of cement products obtained by grinding with different ratios of amine and polymer group grinding chemicals and A2 polymer reveals a transition from positive to negative values. It was observed that when anionic groups of grinding chemicals absorbed onto cement grains, it changes where cationic and anionic groups are located. In polymer groups, it was shown that the particles of the cement product obtained as a result of A2-2.5 g grinding process form a more anionic system compared to other grinding chemicals. As the zeta potential undergoes a negative transition, the repulsion force between the particles is increased, thereby enhancing the grinding performance. This assertion is corroborated by the findings of Blaine and the 32 µm sieve values [30].

In order to elucidate the adsorption behavior, the majority of researchers have proposed a mosaic structure with unevenly distributed positive and negative charges on the cement particle surfaces [7,24,25,26,27,28].

Following the initiation of the hydration reaction, the deposition of fine hydration products such as ettringite (AFt, 100–500 nm) and calcium silicate hydrates (C-S-H, <500 nm) occurs on the solid phases, thereby forming surfaces with high heterogeneity [27,34,35,36,37,38,39,40]. It has been found that this structure remains largely unchanged during the initial hour of hydration [35]. The complexity of the charged mosaic surface is further increased, depending on the chemical composition and ionic strength of the aqueous phase. The zeta potentials of these solid phases are known to develop through the adsorption of anions and cations in the Stern layer, in the vicinity of the particle/solution interface [41]. Multivalent ions, most notably divalent ions such as Ca^2+^ and SO_4_^2−^, which are present in high concentrations within cementitious materials, exhibit a strong propensity for adsorption on the surfaces of cement particles that bear opposing charges. Figure 7 and Figure 8 present the ion chromatography of cement samples.

In the domain of ion chromatography, calcium ions (Ca^2+^) and sulfate ions (SO_4_^2−^) are of particular significance. In the Stern layer, opposite changes in zeta potential are observed due to anion and cation absorption [24,25]. It is evident that these opposite movements result in a significant amount of Ca^2+^ and SO_4_^2−^ binding occurring on the surface of the cement grains. The directional decreasing movement in the zeta potential, indicated by the negative sign, is associated with an increase in SO_4_^2−^ bonds and a decrease in Ca^2+^ bonds. This situation is exemplified by a polymer-based grinding chemical with a concentration of 2.5 g.

These divalent counterions have been observed to reduce the zeta potential, and in some cases, to reverse the charge of these surfaces. In the case of C-S-H, an increase in Ca^2+^ concentration has been shown to increase the zeta potential. The process of charge reversal occurs at a higher concentration of Ca(OH)_2_ when the pH is greater than 12. This is due to the significant accumulation of calcium counterions, which results in the overcompensation of surface charge [32]. For AFt, an increase in the concentration of SO_4_^2−^ has been shown to decrease the zeta potential, with the occurrence of charge reversal being observed at a certain SO_4_^2−^ concentration.

The most abundant cations in cement are Ca^2+^ cations. As illustrated in Figure 8 and Figure 9, the process of bond formation between calcium cations (Ca^+2^) and silicon anions (Si^−2^) leads to a decline in the number of free calcium cations within the system. The presence of calcium and silicon ions (Ca^+2^ and Si^−2^) has been observed to result in the formation of a gel layer surrounding the cement grains. This is particularly evident in the case of C-S-H bonds, where the negative shift in the zeta value to −8.98 mV in the cement product containing the A2-2.5 g additive indicates a decline in the zeta potential. This decline can be attributed to a reduction in the free Ca^+2^ cation concentration within the system. For the ettringite phase, the decrease in zeta potential indicates an increase in the free SO_4_^2−^ anion within the system [38].

The identification of surface changes in cement was facilitated by the presence of zeta and ion changes, which are indicative of anions and cations, respectively. An additional factor that was identified as contributing to the process was water, which was found to be present in grinding aids. It is evident that the high activity of these ions would result in the appearance of water on the interior surface. The analysis focused on the characterization of specific grinding aids, with a particular emphasis on those that exhibited the capacity to remain suspended in an aqueous medium. The results of the turbidity analysis, with varying PCE and A2 grinding aids, are presented in Figure 9 and Figure 10, respectively.

Turbidity measurements were examined over seven distinct periods, including the first minute, five minutes, 15 min, 30 min, one hour, 12 h, and 24 h. In consideration of the research findings, it was determined that the presence of A2 polymer in water led to an increase in turbidity, attributable to its propensity to remain suspended. The following essay will provide a comprehensive overview of the relevant literature on the subject. Following a rigorous examination of the available data, it was determined that the most efficient rate of A2 polymer was 2.5 g. This is since the cement grinding chemical A2, which is designed as a polymer-based grinding chemical, contains a calcium ion in the tail part of the chain structure. This process leads to the formation of SO_4_^2−^ and Ca^+2^ ions, which are present in the system during the hydration reaction. The bond formation between these ions maintains the system’s dynamic equilibrium. Concomitantly, the turbidity present within the solution serves as an indication of the efficacy of polymer-based grinding aids in enhancing the fluidity of concrete. These polymers, which find application in the field of concrete science, contribute to the preservation of consistency and the acceleration of the cement setting process at the initiation of cement hydration, a phenomenon precipitated by the interaction of clinker with water.

Studies on how polymer-based chemical additives react with well-known cements have taught us more about the next generation of chemicals and how to make new, improved products [42]. Polymer admixtures are made up of a main polymer backbone made of carbon atoms. Carboxylic acid and long-chain polyether side groups are attached to this. A shorter polymer backbone and longer/more numerous ether side chains together make it easier and longer lasting to process [43]. As superplasticizer dosage increases, plastic viscosity decreases. At concentrations above a critical level, the flow changes to a Newtonian state, where the attraction between particles is reduced by the additive [15,44]. Viscosity (cP) values with varying PCE and A2 grinding aids are presented in Figure 11 and Figure 12.

Upon examination of Figure 11 and Figure 12, the viscosity values of cement products obtained by grinding with different grinding chemicals generally decrease with increasing use of polymer-based grinding chemicals. It is evident that the repulsion force between cement grains is increased, leading to a reduction in agglomeration and the maintenance of a homogeneous structure within the system. The findings of A2-2.5 g polymer-based grinding chemicals further corroborate this assertion. However, it is imperative to ascertain the optimal usage rate of polymer-based grinding chemicals. When a certain amount of polymers are used up, it is seen that the chain fringes get more tangled and farther away from the cement grains. This situation simultaneously reduces the effectiveness of polymers in applying an effective force on cement particles. In particular, the grinding chemistry of A2-4 g polymer-based systems demonstrates a reversal of the mechanism of action.

## 4. Conclusions

The present study investigated the effects of polymer-based grinding aids, utilized in the context of clinker grinding, on the surface chemistry properties of cement. The evaluated surface properties included zeta potential, ion chromatography, turbidity and rheology of cement.

Following a series of experiments involving the testing of polymer-based grinding chemicals on cement samples, it was determined that chemical A2 is more active in terms of dispersant and agglomeration-preventing surface activity than the other chemicals. Furthermore, it was found to produce a (-) charged zeta value, which is an indication of this activity [31,32,33,34,35,36,37].

As indicated by the zeta values, particularly those exhibiting a positive to negative shift, it was imperative to undertake a thorough investigation into the alterations in the ion composition of the surface chemistry of cement.

The zeta potential of the cement sample that has undergone a grinding process with the A2-2.5 g polymer-based aid was −8.98 mV. In response to this zeta value, the amount of Ca^+2^ binding decreased to 190 mg/L and the amount of SO_4_^−2^ binding increased to 400 mg/L. It was observed that ion changes were more prevalent in the A2-2.5 g group.

It has been observed that, in the case of C-S-H, an increase in the Ca^2+^ concentration results in an increase in the zeta potential. The process of charge reversal occurs at a higher concentration of Ca(OH)_2_ when the pH is greater than 12. This is due to the significant accumulation of calcium counterions, which results in the overcompensation of surface charge [35]. For AFt, an increase in the SO_4_^−2^ concentration has been shown to decrease the zeta potential, with the charge reversal occurring at a certain SO_4_^−2^ concentration.

Turbidity tests showed that a cement sample ground with 2.5 g of A2 polymer remained suspended for 24 h. The hypothesis that the consistency of concrete produced with this cement would be maintained for an extended period is supported by the observation that the concrete remained homogeneous for a considerable time.

Rheological measurements showed that plastic viscosity decreased with increasing use of polymer-based grinding chemicals. At the optimum dosage concentration, the flow transitioned to a Newtonian flow state by reducing the attraction between the additive and the cement particles [15,44]. This was thought to be due to increased repulsive forces between cement particles, reduced agglomeration, and the maintenance of a homogeneous structure within the system. The rheological results with 2–2.5 g of polymer-based grinding chemicals also confirm this claim.

In comparison with conventional amine-group grinding chemicals, the utilization of the synthesized A2 polymer at the optimal concentration (2.5 g) has the potential to yield environmental benefits by reducing both polymer and grinding energy costs, as well as producing up to a 10% increase in strength. The following article will provide a comprehensive overview of the relevant literature [30].

In addition, a new article on the effects of these new-generation grinding additives on cement hydrophobicity and rheology has been prepared and submitted to this journal (*MDPI Polymers*).

The findings of this study indicate that the synthesis of polymer structures can be an effective method for reducing agglomeration in the context of cement production and thus facilitating the grinding process [30]. Furthermore, turbidity tests are believed to indicate the range within which the setting time can be stabilized at the point where cement hydration begins.

## Figures and Tables

**Figure 1 polymers-17-02691-f001:**
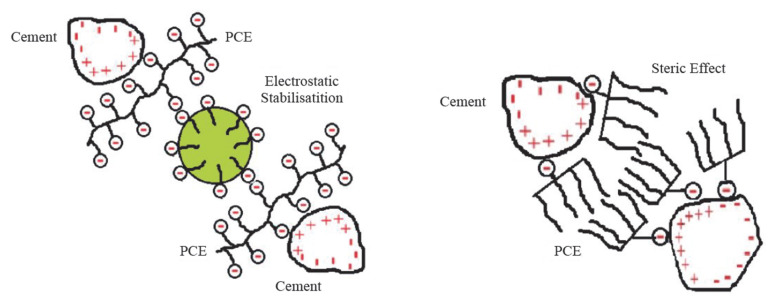
Electrostatic and steric effect between PCE and cement grains.

**Figure 2 polymers-17-02691-f002:**
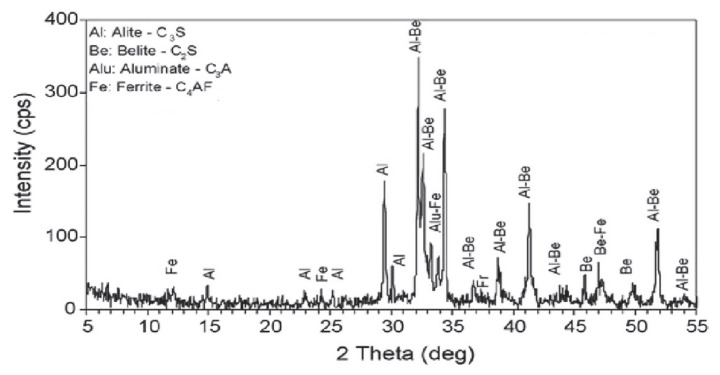
XRD analysis of clinker.

**Figure 3 polymers-17-02691-f003:**
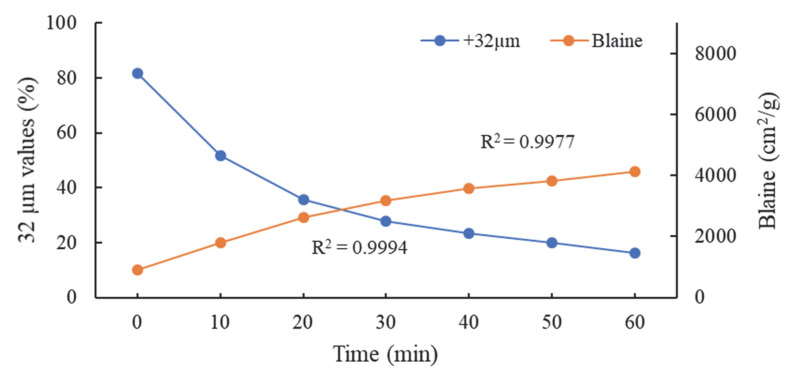
32 µm (%) sieve residue and Blaine (cm^2^/g) values of clinker.

**Figure 4 polymers-17-02691-f004:**
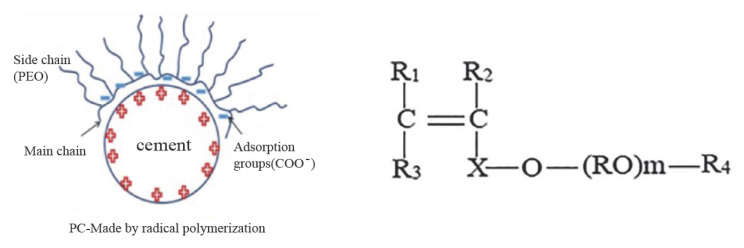
PCE made by radical polymerization (**left**)—General formula of A1-A2, and A3 chemical additives (**right**).

**Figure 5 polymers-17-02691-f005:**
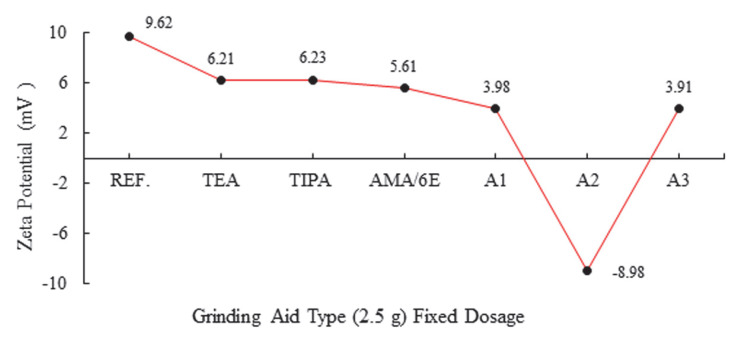
Effect of different grinding aid types with fixed dosage (2.5 g) on zeta potential.

**Figure 6 polymers-17-02691-f006:**
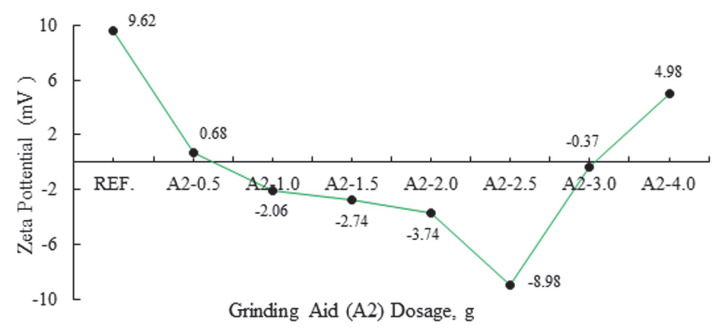
Effect of varying dosages of A2 grinding aid on zeta potential.

**Figure 7 polymers-17-02691-f007:**
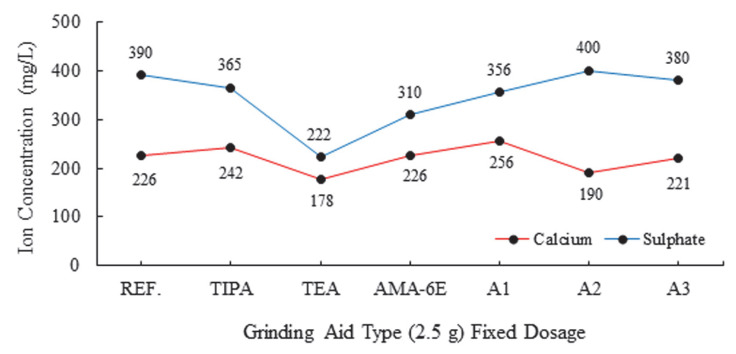
Effect of different grinding aid types with fixed dosage (2.5 g) on ion concentrations.

**Figure 8 polymers-17-02691-f008:**
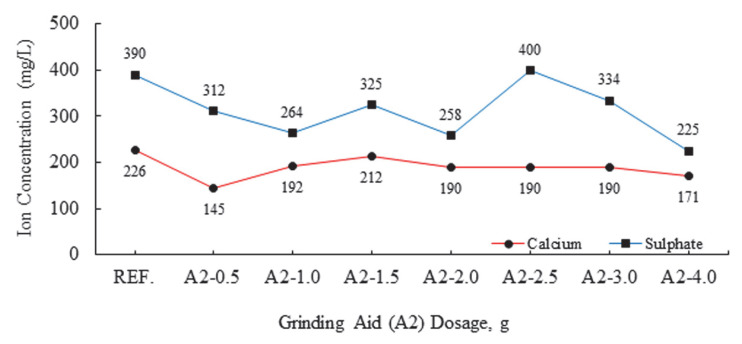
Effect of varying dosage of the A2 grinding chemical on ion concentrations.

**Figure 9 polymers-17-02691-f009:**
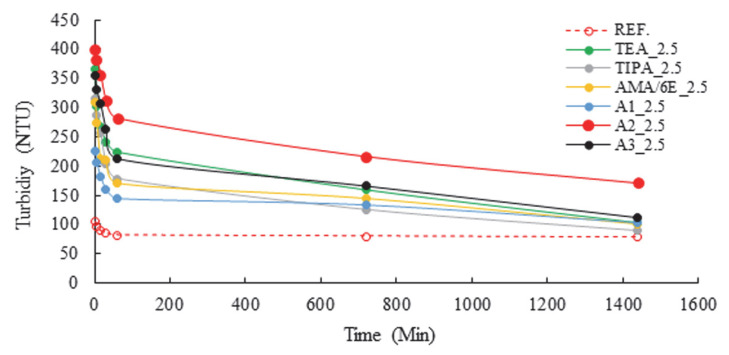
Turbidity data of different polymer-based grinding aids at a fixed concentration (2.5 g).

**Figure 10 polymers-17-02691-f010:**
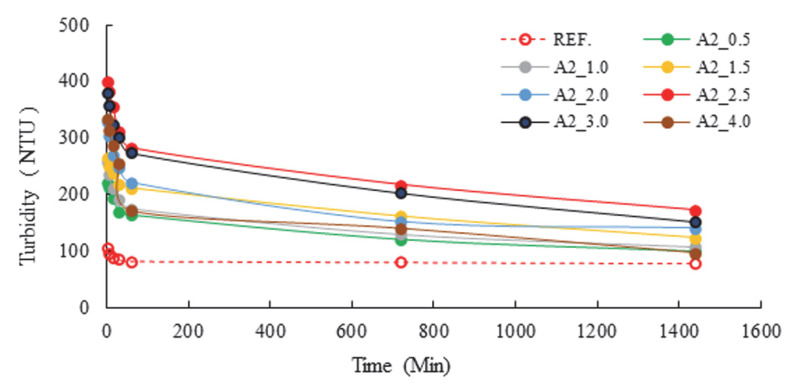
Turbidity data of varying concentrations of the A2 polymer-based grinding aid.

**Figure 11 polymers-17-02691-f011:**
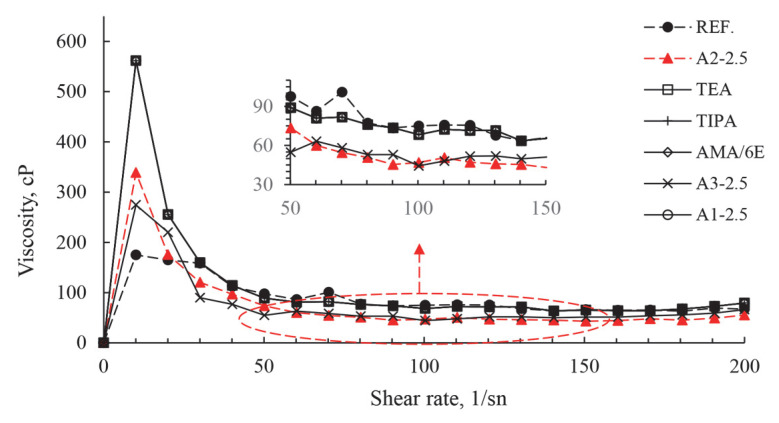
Effect of different grinding aids at a fixed concentration (2.5 g) on cement viscosity.

**Figure 12 polymers-17-02691-f012:**
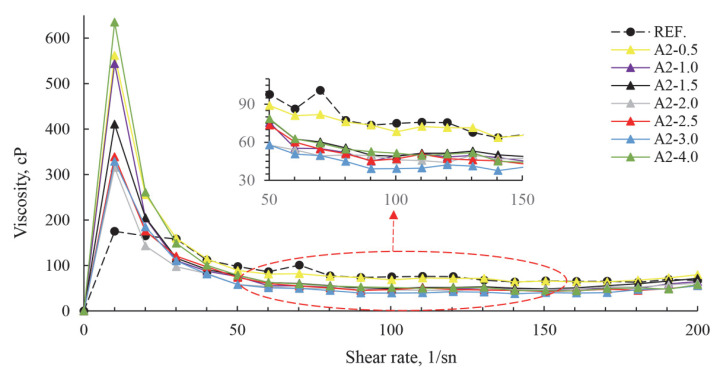
Effect of varying concentrations of A2 grinding aid on cement viscosity.

**Table 1 polymers-17-02691-t001:** The chemical composition of the clinker, gypsum, and limestone.

Major Oxides	Clinker (%)	Gypsum (%)	Limestone (%)
SiO_2_	20.62	1.45	0.00
Al_2_O_3_	4.95	0.54	0.15
Fe_2_O_3_	3.87	0.25	0.10
CaO	65.41	34.27	56.25
MgO	1.93	0.33	0.47
SO_3_	0.63	40.83	0.04
LOI	0.41	22.22	42.67
Na_2_O	0.30	0.03	0.03
K_2_O	0.73	0.11	0.02
TiO_2_	0.30	0.00	0.00
Mn_3_O_4_	0.23	0.00	0.01
Total	99.38	100.03	99.74

**Table 2 polymers-17-02691-t002:** The physical and chemical properties of the grinding aids.

Polymer	Solid Rate (%)	pH	Color	Density (g/cm^3^)	Molecular Weight (g/mol)
TEA	88	11.6	Transparent	1.12	149.19
TIPA	78	7.8	Transparent	1.02	191.27
AMA-6E	52	5.6	Brown	1.07	58,700
A1	50	5.8	Light Yellow	1.05–1.15	59,000
A2	50	5.73	Transparent	1.05–1.15	60,000
A3	50	6.0	Transparent	1.05–1.15	58,900

**Table 3 polymers-17-02691-t003:** Cement samples that analyzed with corresponding instruments.

Analyze Name	Instrument
Zeta Potential	NanoBrook ZetaPALS
Ion Chromatography	Thermo Dionex ICS-1100
Turbidity	Thermo Orion AQUAfast II AQ2010
Viscosity	Brookfıeld R/S Plus Rheometer

## Data Availability

The original contributions presented in this study are included in the article. Further inquiries can be directed to the corresponding author.

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
