# Peer review of "Investigation of the Effects of Polymer-Based Grinding Aids on the Surface Chemistry Properties of Cement"

_polymers, 2025, doi:10.3390/polym17192691_

Round 1
Reviewer 1 Report
Comments and Suggestions for Authors
The manuscript presents an experimental study on the influence of polymer-based grinding aids, with emphasis on a synthesized polymer (A2), on the surface chemistry of Portland cement. The study addresses a topic of practical importance in cement chemistry and concrete technology, with potential implications for the development of more efficient grinding aids. The manuscript is generally well-structured, and the experimental program is clearly described. The work demonstrates novelty in focusing on polymer-based grinding aids synthesized in the laboratory rather than only relying on conventional amine-based chemicals. However, several issues regarding clarity, data interpretation, and contextualization within the broader literature should be addressed.
- Several sections (especially Introduction and Results) contain long, repetitive explanations of superplasticizer mechanisms. These should be condensed for clarity.
- The text occasionally includes tangential discussions (e.g., on surfactant head groups) that could be shortened or refocused on cement-related implications.
- Figures (e.g., Figs. 6–14) need clearer captions and labels. For example, axes should include units consistently (mV, mg/L, NTU, cP). Some legends are difficult to interpret, and the figures require higher resolution for readability.
- The discussion sometimes overstates findings without sufficient quantitative backing. For instance, the claim that turbidity directly indicates concrete consistency over time is not fully substantiated. More cautious interpretation is recommended.
- The manuscript does not mention whether experiments were repeated or how reproducibility was ensured. Without statistical treatment (standard deviations, error bars), it is difficult to assess the robustness of the findings.
- Sample size (n = 13) is mentioned, but variation and confidence in results are not adequately addressed.
- The discussion should better highlight how the present findings advance beyond prior studies rather than reiterating well-known superplasticizer mechanisms.
- The conclusions section is somewhat descriptive and repeats results rather than synthesizing broader implications. It should more clearly state the novelty of A2 compared to traditional grinding aids; practical recommendations for dosage; and limitations of the study and directions for future research.
- Figure 8 should be deleted. It is common sense.
Author Response
Dear reviewer, we would like to thank you sincerely for your scientific contribution and valuable comments on our manuscript. I've included your comments and our responses below.
Best regards.
Dr. Ebru Dengiz ÖZCAN
Comments 1: [Several sections (especially Introduction and Results) contain long, repetitive explanations of superplasticizer mechanisms. These should be condensed for clarity.]
Response 1: All required corrections have been performed. The introduction and conclusion sections have been rewritten and marked in red.
Comments 2: [The text occasionally includes tangential discussions (e.g., on surfactant head groups) that could be shortened or refocused on cement-related implications. ]
Response 2: In line 83 there is a surfactant head group information but also there was another one in 101. It is deleted and changed a shortly brief.
Comments 3: [Figures (e.g., Figs. 6–14) need clearer captions and labels. For example, axes should include units consistently (mV, mg/L, NTU, cP). Some legends are difficult to interpret, and the figures require higher resolution for readability. ]
Response 3: The graphical representation of the Zeta Potential data (lines 260-264), the ion concentration data (lines 299-306), the turbidity data (lines 340-346) and the viscosity data (lines 370-378) was modified.
Comments 4: [The discussion sometimes overstates findings without sufficient quantitative backing. For instance, the claim that turbidity directly indicates concrete consistency over time is not fully substantiated. More cautious interpretation is recommended. ]
Response 4: The discussion section has been rewritten with more cautious comments based on your suggestions.
Comments 5: [The manuscript does not mention whether experiments were repeated or how reproducibility was ensured. Without statistical treatment (standard deviations, error bars), it is difficult to assess the robustness of the findings. ]
Response 5: It is mentioned in line 174-177.
Comments 6: [Sample size (n = 13) is mentioned, but variation and confidence in results are not adequately addressed. ]
Response 6: In the Result and Discussion section, line 247-255, it is explained and also given the tests which are dealed with.
Comments 7: [The discussion should better highlight how the present findings advance beyond prior studies rather than reiterating well-known superplasticizer mechanisms. ]
Response 7: Highlight explanations are rewritten and marked in red.
Comments 8: [The conclusions section is somewhat descriptive and repeats results rather than synthesizing broader implications. It should more clearly state the novelty of A2 compared to traditional grinding aids; practical recommendations for dosage; and limitations of the study and directions for future research. ]
Response 8: In line 463-511 Conclusions section is rewritten and given more highlights.
Comments 9: [Figure 8 should be deleted. It is common sense. ]
Response 9: The figure 8 is deleted and given small explanation in the paragraph line 320-338.
Reviewer 2 Report
Comments and Suggestions for Authors
- In the conclusion section of the abstract, the statement "A2 exhibited superior dispersant activity" lacks data support. While the performance advantage of A2 is attributed to its surface activity, critical evidence regarding molecular adsorption is not provided. It is recommended to supplement the analysis of the key data point, the zeta potential value of -8.98 mV, and cite highly relevant literature to corroborate the proposed mechanism.
- The testing conditions for zeta potential have not been standardized, as the ionic strength and pH of the solution were not controlled during the measurement. This raises doubts about the comparability of the data. It is recommended to supplement relevant control experiments and error analysis.
- The description of differences in charge properties among the molecular population of cement particles is overly vague, and the charge distribution model is excessively simplified, lacking support from specific models.
- The rheological state experiments are not correlated with engineering performance. The laboratory viscosity data (Figures 13-14) have not been linked to the macroscopic properties of concrete. It is recommended to supplement correlation experiments between cement paste and concrete, such as tests on slump and compressive strength.
- The evidence for attributing ionic changes is incomplete. Although the increase in SO₄²⁻ is attributed to the effect of A2, potential influences such as accelerated gypsum dissolution have not been ruled out. It is recommended to supplement quantitative analyses at different hydration stages.
- The basis for optimizing the dosage of A2 is insufficient. The selection of the A2 dosage (A2-2.5g) is solely based on experimental data, without explaining the reason for the non-linear response (e.g., performance degradation at A2-4g). Relevant curve analysis should be added to address this issue.
- Conclusions such as "the negative shift of zeta potential leads to increased SO₄²⁻ binding" are overly simplified, as other influencing factors (e.g., pH variation) have not been excluded. Supplementary control experiments should be added to provide supporting evidence for this conclusion.
- Abbreviations in the reference list are inconsistently used (e.g., "Construct. Build. Mater." and "Constr. Build. Mater." are mixed), and the proportion of literatures published in the past three years is relatively low. Additionally, relevant original studies have not been cited for key theoretical models.
- There are some grammatical errors in the text. Additionally, the zeta mechanism is repeatedly discussed in multiple chapters, making it overly redundant. It is recommended to refine the language structure for conciseness.
Author Response
Dear reviewer, we would like to thank you sincerely for your scientific contribution and valuable comments on our manuscript. I've included your comments and our responses below.
Best regards.
Dr. Ebru Dengiz ÖZCAN
Comments 1: In the conclusion section of the abstract, the statement "A2 exhibited superior dispersant activity" lacks data support. While the performance advantage of A2 is attributed to its surface activity, critical evidence regarding molecular adsorption is not provided. It is recommended to supplement the analysis of the key data point, the zeta potential value of -8.98 mV, and cite highly relevant literature to corroborate the proposed mechanism.
Response 1: Our second article submitted to the same journal includes additional measurements such as FT-IR and SEM images. Therefore, we did not wish to go into further detail in this article. The Blaine and turbidity values in this article, along with the slump test data to be included in our other article, demonstrate the dispersing and fluidizing properties of polymer-based chemical additives. Additionally, the abstract section of the manuscript has been rewritten according to your suggestions.
Comments 2: The testing conditions for zeta potential have not been standardized, as the ionic strength and pH of the solution were not controlled during the measurement. This raises doubts about the comparability of the data. It is recommended to supplement relevant control experiments and error analysis.
Response 2: Zeta potential measurements were conducted using pure water at its natural pH in climate-controlled laboratory conditions. Because cement is a heterogenic structure rather than a single mineral, ionic strength and pH control were not standardized. Therefore, the natural pH and ions released from the cement were analyzed as is. Ion concentration data are also included in the study.
Comments 3: The description of differences in charge properties among the molecular population of cement particles is overly vague, and the charge distribution model is excessively simplified, lacking support from specific models.
Response 3: While we agree with your valuable opinion, we had to refrain from making bold and definitive comments, considering the suggestions of other reviewers. We thank you for your understanding on this matter.
Comments 4: The rheological state experiments are not correlated with engineering performance. The laboratory viscosity data (Figures 13-14) have not been linked to the macroscopic properties of concrete. It is recommended to supplement correlation experiments between cement paste and concrete, such as tests on slump and compressive strength.
Response 4: We also shared the cement and concrete strength in the first part of our article which was published in a polymer journal. It was referenced as number [34].
Comments 5: The evidence for attributing ionic changes is incomplete. Although the increase in SO₄²⁻ is attributed to the effect of A2, potential influences such as accelerated gypsum dissolution have not been ruled out. It is recommended to supplement quantitative analyses at different hydration stages.
Response 5: The clinker used in both the reference and blended cement samples is completely the same. Therefore, numerical analysis was not performed at different hydration stages.
Comments 6: The basis for optimizing the dosage of A2 is insufficient. The selection of the A2 dosage (A2-2.5g) is solely based on experimental data, without explaining the reason for the non-linear response (e.g., performance degradation at A2-4g). Relevant curve analysis should be added to address this issue.
Response 6: In preliminary research for the manuscript, different polymer-based grinding chemicals and amine-group grinding chemicals were compared in another polymer journal. Compressive strength and Blaine tests determined that A2 had the most effective performance. Subsequently, tests were conducted in the 0,5-4 g range to determine the optimum use rate of A2 for this manuscript. The optimum use rate was determined based on these data. The data, which is particularly noteworthy and supported by literature studies on concrete slump properties, demonstrates that using chemical admixtures above the optimum use rate can increase agglomeration and cause concrete slump. This slump data is also described in a second manuscript published in the same journal.
Comments 7: Conclusions such as "the negative shift of zeta potential leads to increased SO₄²⁻ binding" are overly simplified, as other influencing factors (e.g., pH variation) have not been excluded. Supplementary control experiments should be added to provide supporting evidence for this conclusion.
Response 7: Since zeta and ion concentration were tested and there were analyses that supported each other in our literature research, no additional tests were needed.
Comments 8: Abbreviations in the reference list are inconsistently used (e.g., "Construct. Build. Mater." and "Constr. Build. Mater." are mixed), and the proportion of literatures published in the past three years is relatively low. Additionally, relevant original studies have not been cited for key theoretical models.
Response 8: All abbreviations have been carefully corrected according to your feedback.
Comments 9: There are some grammatical errors in the text. Additionally, the zeta mechanism is repeatedly discussed in multiple chapters, making it overly redundant. It is recommended to refine the language structure for conciseness.
Response 9: Grammatical errors found in the manuscript have been corrected. Repetitions of information have been eliminated.
Reviewer 3 Report
Comments and Suggestions for Authors
General comment
This work investigates the influence of polymer-based grinding aids on the surface properties of cement materials. The authors conducted various tests to assess parameters such as zeta potential, turbidity, and rheological behavior. The study aligns with the journal’s aims and scope and has the potential for acceptance pending the implementation of the following revisions.
Reviews required
- Line 107: Please provide a reference for EN 197-1 to support the standards mentioned.
- Figures 4 and 5: These can be combined into a single figure with subfigures to improve clarity and conserve space.
- Section 2.3: Including a summary table listing all tested specimens alongside their respective test types would facilitate clearer understanding of the experimental design.
- Lines 269–276: The phrases “it has been established” and “it has been demonstrated” are used repeatedly without supporting references. Please add appropriate citations or rephrase to avoid unsupported assertions.
- Figure 8: The figure is embedded within the text but not referenced elsewhere. Ensure it is appropriately cited and discussed in the text.
- Line 289: The phrase “it has been demonstrated” appears multiple times in the manuscript. To maintain a professional and polished writing style, avoid excessive repetition of this expression.
- Figures 6 to 9: The use of 3D projections to display 2D data may be inappropriate. Consider using simpler 2D plots for clearer visualization of the results.
- Lines 307–313: The font size in this section appears inconsistent with the rest of the document. Please standardize the font size throughout the manuscript, including lines 347–370 and 415–418, where discrepancies are also observed.
- To enhance the presentation of results, consider including additional photographs or images of the experimental procedures, which would strengthen the overall depiction of the study.

Author Response
Dear reviewer, we would like to thank you sincerely for your scientific contribution and valuable comments on our manuscript. I've included your comments and our responses below.
Best regards.
Dr. Ebru Dengiz ÖZCAN
Comments 1: [Line 107: Please provide a reference for EN 197-1 to support the standards mentioned. ]
Response 1: EN 197-1 Standard Reference is given as number [34].
Comments 2: [Figures 4 and 5: These can be combined into a single figure with subfigures to improve clarity and conserve space. ]
Response 2: Figure 4 and 5 are combined and named as Table 4 in line 173-174.
Comments 3: [Section 2.3: Including a summary table listing all tested specimens alongside their respective test types would facilitate clearer understanding of the experimental design. ]
Response 3: The code of the cement samples are given in line 248-250. Also, tests are listed in Table 4 in line 253-254.
Comments 4: [Lines 269–276: The phrases “it has been established” and “it has been demonstrated” are used repeatedly without supporting references. Please add appropriate citations or rephrase to avoid unsupported assertions. ]
Response 4: In line 13-15, 24-26, 28-30, 146, 296, 299, 322, 326, 41-414, 454-459, 487 and 499, ‘established’ and ‘demonstrated’ words are changed with another same meaning.
Comments 5: [Figure 8: The figure is embedded within the text but not referenced elsewhere. Ensure it is appropriately cited and discussed in the text. ]
Response 5: Another reviewer wants to delete it. Because it was a general information. I gave a little information in the paragraph.
Comments 6: [Line 289: The phrase “it has been demonstrated” appears multiple times in the manuscript. To maintain a professional and polished writing style, avoid excessive repetition of this expression. ]
Response 6: Repetition of the same expressions has been avoided. In line 13-15, 24-26, 28-30, 146, 296, 299, 322, 326, 411-414, 454-459, 487 and 499, ‘established’ and ‘demonstrated’ words are changed.
Comments 7: [Figures 6 to 9: The use of 3D projections to display 2D data may be inappropriate. Consider using simpler 2D plots for clearer visualization of the results. ]
Response 7: The figures 6-7-8-9 are redrawn.
Comments 8: [Lines 307–313: The font size in this section appears inconsistent with the rest of the document. Please standardize the font size throughout the manuscript, including lines 347–370 and 415–418, where discrepancies are also observed. ]
Response 8: All the manuscript font size and also figure and table’s styles were checked.
Comments 9: [To enhance the presentation of results, consider including additional photographs or images of the experimental procedures, which would strengthen the overall depiction of the study. ]
Response 9: All revisions have been made in accordance with your comments. However, new experimental data and images could not be added because a new manuscript has been submitted to the this journal. Thank you for your understanding on this matter.
Reviewer 4 Report
Comments and Suggestions for Authors
- Abstract is verbose without any focus. It is suggested to rewrite it the focus on novelty, methodology adopted, key outcomes and suggestions in a brief way.
- Novelty of the work should be clearly stated with an emphasize on significant advancement in the existing literatures beyond past investigations on polymer-based grinding aids and PCEs.
- Introduction is overloaded with fundamentals of superplasticizer without any critical analysis on the research gap pertaining to polymer-based grinding aids.
- Polymer synthesis is not deeply discussed. To ensure reproducibility, characterization, purity checks and reaction conditions may be deeply discussed.
- The dosage adopted for A2 polymer of 0.5-4g is not properly justified. Is it based on past literature or trials or practical relevance?
- Ther is a strong mechanistic claim on hydration process and ion exchange, but the data doesn’t realistically support these claims.
- The section on turbidity is weak with the results presented descriptively without any quantitative comparison or statistical treatment or explanation on mechanism apart from simple suspension stability.
- The claims on formation of C-S-H or ettringite or gel layers are hypothetical without any SEM/XRD/FTIR evidence after chemical treatment.
- Several sections provide repetitive general explanations without offering innovative visions on PCEs, hydration, electrostatic repulsion and surfactant adsorption.
- There is a claim on industrial and performance benefits in the conclusion section, whereas the study is limited to laboratory-scale measurements without any link with the real performance of cement/concrete.
- The study lacks environmental impacts, costs or scalability of the proposed grinding aids with the conventional ones, leading to practical limitations.
- Quality of figures to be improved (Ex. Figure 2, 4). Figure 3 to be checked for Labelling on both curves.
- Be consistent with the units. For ex. Somewhere it is mentioned as ‘mg/l’ and somewhere as ‘mg/L’. Similarly, ‘gr’ and ‘g’, ‘cm²/g’ and ‘cm²/gr’.
- There is a clustering of presenting data with inconsistent scales in Figs. 9 to 12 without any discussion on the significance of differences.
Author Response
Dear reviewer, we would like to thank you sincerely for your scientific contribution and valuable comments on our manuscript. I've included your comments and our responses below.
Best regards.
Dr. Ebru Dengiz ÖZCAN
Comments 1: [Abstract is verbose without any focus. It is suggested to rewrite it the focus on novelty, methodology adopted, key outcomes and suggestions in a brief way. ]
Response 1: The abstract has been rewritten according to your suggestions.
Comments 2: [Novelty of the work should be clearly stated with an emphasize on significant advancement in the existing literatures beyond past investigations on polymer-based grinding aids and PCEs. ]
Response 2: Auxiliary chemicals used in the cement grinding process are described in the literature as amine-based and general polymer raw materials. However, auxiliary chemicals used as new generation grinding aids through polymer synthesis have not yet been included in the literature. We believe that our study will be among the first to be published on this subject.
Comments 3: [Introduction is overloaded with fundamentals of superplasticizer without any critical analysis on the research gap pertaining to polymer-based grinding aids. ]
Response 3: The introduction section has been revised in line with your suggestions.
Comments 4: [Polymer synthesis is not deeply discussed. To ensure reproducibility, characterization, purity checks and reaction conditions may be deeply discussed. ]
Response 4: This article is the second part of our investigation. Our first part of article was published in a polymer journal which we explained the details about its production line. Also I referenced it in the references number [34].
Comments 5: [The dosage adopted for A2 polymer of 0.5-4g is not properly justified. Is it based on past literature or trials or practical relevance? ]
Response 5: In this study, pure polymer obtained through synthesis was used. Dosage intervals were selected based on the chemical structure, composition, and molecular weight of the polymer, as well as information obtained in our first article. We believe this study will contribute to the literature.
Comments 6: [There is a strong mechanistic claim on hydration process and ion exchange, but the data doesn’t realistically support these claims. ]
Response 6: Figure 8 in the manuscript has been deleted and partial wording changes have been made.
Comments 7: [ The section on turbidity is weak with the results presented descriptively without any quantitative comparison or statistical treatment or explanation on mechanism apart from simple suspension stability. ]
Response 7: This section has been revised again, shortened in accordance with your and other reviewers' opinions, so that it does not contain any bold statements.
Comments 8: [ The claims on formation of C-S-H or ettringite or gel layers are hypothetical without any SEM/XRD/FTIR evidence after chemical treatment. ]
Response 8: In our new article which have sent in this journal has FT-IR and SEM analyses.
Comments 9: [ Several sections provide repetitive general explanations without offering innovative visions on PCEs, hydration, electrostatic repulsion and surfactant adsorption. ]
Response 9: The view of the manuscript is changed without repeated informations.
Comments 10: [ There is a claim on industrial and performance benefits in the conclusion section, whereas the study is limited to laboratory-scale measurements without any link with the real performance of cement/concrete. ]
Response 10: This new generation polymer-based grinding chemical, published in another polymer journal, has been tested for strength, blain, 32 µm, and specific gravity on cement and concrete. Furthermore, the results obtained were also tested industrially at the After-Sales Quality Monitoring unit of Nuh Cement factory. Consistent results were obtained in both laboratory and industrial environments.
Comments 11: [ The study lacks environmental impacts, costs or scalability of the proposed grinding aids with the conventional ones, leading to practical limitations. ]
Response 11: We also tested this new polymer-based grinding aid ( A2 ) in Nuh Cement Factory in TURKEY, but this is another challenge and also we had a very successful result. If you would like to see them, we can share with you.
Comments 12: [ Quality of figures to be improved (Ex. Figure 2, 4). Figure 3 to be checked for Labelling on both curves. ]
Response 12: All figures was checked and changed.
Comments 13: [ Be consistent with the units. For ex. Somewhere it is mentioned as ‘mg/l’ and somewhere as ‘mg/L’. Similarly, ‘gr’ and ‘g’, ‘cm²/g’ and ‘cm²/gr’. ]
Response 13: All symbols are checked and changed.
Comments 14: [ There is a clustering of presenting data with inconsistent scales in Figs. 9 to 12 without any discussion on the significance of differences. ]
Response 14: The scaling of the turbidity and viscosity graphs has been redone. More explicit statements regarding the graphs have been shared.
Round 2
Reviewer 1 Report
Comments and Suggestions for Authors
The basic information of Polymer Based Grinding Aids should be given. Its performance should be compared with ones of
existing aids.
Author Response
Reviewer 1. Second Round
Comments: The basic information on Polymer-Based Grinding Aids should be given. Its performance should be compared with that of existing aids.
Response: For the polymer-based grinding aids, their working mechanisms were explained in lines 40-51. PCEs to cement particles disperses into smaller agglomerates, which become the predominant entities of the cement paste in the concrete mixture. This dispersion has a significant impact on the fluidity and workability of concrete. The development of a negative charge on the cement particles develops as a consequence of this process [5], [6]. Van der Waals forces, which appear to control agglomeration, can be neutralized by anionic polymers adsorbed onto the particles, as well as by sulphonic groups on their surface [7]. Cement particle dispersion can be attributed to electrostatic repulsion arising from the adsorption of negatively charged groups. These forces are chiefly contingent on the extent to which PCEs are adsorbed onto the cement particle surface [8]. Consequently, the magnitude of PCE adsorption on the cement particle surface directly correlates with the magnitude of repulsive forces between particles, thereby affecting the fluidity of the cement paste [9]. Therefore, an understanding of the behavior of cement particles can clarify the interaction between cement and PCEs, which in turn helps elucidate the adsorption behavior [10-14].
PCEs that were using for grinding aids synthesis details were given in lines 146-148.
Vinylpolyethyleneglycol (VPEG) was synthesized using acrylic acid, a copolymer. This is an addition reaction initiated by a free radical starting chemical. Azobis (C8H12N4) was used as an initiator. It works at low temperatures. It is an organic solvent.
Grinding aids’ performance were compared to other existing aids in line 452-456.
In comparison with conventional amine-group grinding chemicals, the utilization of the synthesized A2 polymer at the optimal concentration (2,5 g) has the potential to yield environmental benefits by reducing both polymer and grinding energy costs, as well as up to 10 % increase in strength. The following article will provide a comprehensive overview of the relevant literature [34].
Thank you for your valuable feedback and support. We believe that, with your help, our manuscript will be successful as a research article.
Reviewer 2 Report
Comments and Suggestions for Authors
All the comments have been addressed.
Author Response
Reviewer 2 - Second Round:
Comments: All the comments have been addressed.
Response: Thank you for your valuable feedback and support. We believe that, with your help, our manuscript will be successful as a research article.
Reviewer 4 Report
Comments and Suggestions for Authors
Appreciation for the authors for their careful addressing of the comments
Author Response
Reviewer 4 - Second Round:
Comments: All the comments have been addressed.
Response: Thank you for your valuable feedback and support. We believe that, with your help, our manuscript will be successful as a research article.
Round 3
Reviewer 1 Report
Comments and Suggestions for Authors
it can be accepted.